# BRIDGING RECOMMENDATION AND MARKETING VIA RECURRENT INTENSITY MODELING

**Yifei Ma, Ge Liu & Anoop Deoras**
AWS AI Labs
{yifeim,gliua,adeoras}@amazon.com

## ABSTRACT

This paper studies some under-explored connections between personalized recommendation and marketing systems. Obviously, these two systems are different, in two main ways. Firstly, personalized item-recommendation (ItemRec) is user-centric, whereas marketing recommends the best user-state segments (UserRec) on behalf of its item providers. (We treat different temporal states of the same user as separate marketing opportunities.) To overcome this difference, we realize a novel connection to Marked-Temporal Point Processes (MTPPs), where we view both problems as different projections from a unified temporal intensity model for all user-item pairs. Correspondingly, we derive Recurrent Intensity Models (RIMs) to extend from recurrent ItemRec models with minimal changes. The second difference between recommendation and marketing is in the temporal domains where they operate. While recommendation demands immediate responses in real-time, marketing campaigns are often long-term, setting goals to cover a given percentage of all opportunities for a given item in a given period of time. We formulate both considerations into a constrained optimization problem we call online match (OnlnMtch) and derive a solution we call Dual algorithm. Simply put, Dual modifies the real-time ItemRec scores such that the marketing constraints can be met with least compromises in user-centric utilities. Finally, our connections between recommendation and marketing may lead to novel applications. We run experiments where we use marketing as an alternative to cold-start item exploration, by setting a minimal-exposure constraint for every item in the audience base. Our experiments are available at https://github.com/awslabs/recurrent-intensity-model-experiments

## 1 INTRODUCTION

Many ML applications today involve decision making based on streams of events. In recommender systems (RecSys), reasoning on the stream of events in a user's past history has allowed real-time user-based item recommendations (ItemRec), leading to significant impacts (Hidasi et al., 2016; Hidasi & Karatzoglou, 2018; Ma et al., 2020; Chen et al., 2019; Kang & McAuley, 2018). However, being user-centric, ItemRec has limitations in creating a globally inclusive environment for the item providers. For example, ItemRec tends to over-expose a few popular items to improve the immediate user satisfaction, yet this causes the selected items to get even more popular, discouraging the other providers from creating seminal items with higher potential values in the future. This is an important limitation because, as a RecSys involves the participation of both users and item providers, we must consider the success of the item providers for the long-term richness of the RecSys environment (Singh & Joachims, 2018; Mladenov et al., 2020; Su et al., 2021).

To cultivate the seed users for the new items, marketing via sponsored advertisements has been a practical strategy. Note that we do not dismiss the possibility of uncontrolled marketing which hurts the utility of the users, but we argue that a minimal amount of controlled marketing in a trusted environment may actually yield better trade-offs between recommendation relevance and item-diversity. For example, we may assign each item provider an exploration budget so that their items get exposed to a minimal percentage of users in a target period of time. (We treat two temporal states of the same user as two user(-state)s with separate marketing opportunities.) We further integrate marketing into recommender systems to create an online matching (OnlnMtch) environment. Notice that Onln-

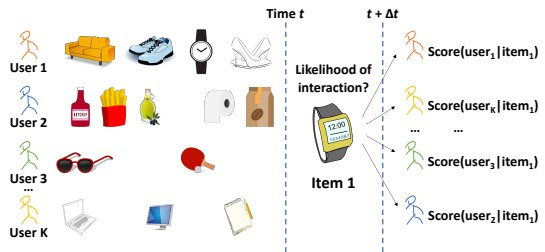 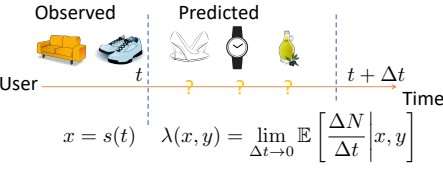

$$x = s(t) \qquad \lambda(x,y) = \lim_{\Delta t \to 0} \mathbb{E}\left[\frac{\Delta N}{\Delta t}\bigg|x,y\right]$$

(a) Definition of the intensity model $\lambda(x,y)$ between User-State $x$ and Item $y$.

Figure 1: UserRec for an item. Users are considered positive if they indeed interact with the given item in a future time window. A matching hit requires both relevance and activeness of the user. An active user may have multiple hits with different items, whereas an inactive user may have zero. In addition, the user hits in a given item are often correlated with their past histories with similar items.

Mtch implies a systemic plan for all user-item recommendations based on the explicit exploration constraints. This is different from implicit uncertainty-driven exploration such as contextual bandits (Li et al., 2010) and Thompson sampling (Agrawal & Goyal, 2013; Kawale et al., 2015). Specifically, the effects of OnlnMtch can be tested immediately using offline datasets, shown in Section 5, whereas the benchmarks for the bandit algorithms often require online updates or their simulations.

To make it happen, we contribute two algorithms: a *RIM* solution to the *UserRec* marketing problem on behalf of the item providers and a *Dual* algorithm for the OnlnMtch problem with its planning challenge. Both are realistic problems with their own applications, which we describe now.

We start with marketing predictions, where we aim to *predict* the relevance of each user with respect to the given item. We validate the quality of our prediction by comparing our top-user recommendations (UserRec) with the set of users who truly interacted with the given item in the holdout back-test period. See Figure 1 for illustration. Besides direct ranking outputs, the raw prediction scores are also used in the OnlnMtch step.

UserRec can be seen as an extension from targeted advertising, where the advertisers bid on search queries or related items to increase the exposure of their chosen items (Chang et al., 2020). In our work, a search query or a related item may be seen as a single event – often the last event – in a user's history. By extending to the full histories of the actual users, we obviously build better user features, but also create a novel problem in the space of information retrieval, which is to compare the users as random time series instead of random variables. For example, the time series data allow us to compare between a relevant but less active user and an active but less relevant user. This would not be feasible with probability density estimation based on the users' hidden-state variables learned for next-item recommendation. See "Bayesian paradox" in Figure 2 for more details.

Our key observation here is in the implicit temporal sensitivity in marketing. The true goal for marketing is to find the users who are most likely to interact with the marketed item in the *next period of time*. This objective can be formally defined by the *intensity parameter* in temporal point processes (TPPs) (Reinhart, 2018; Rizoiu et al., 2017). See Figure 1a for more details. We further use a Marked-TPP decomposition, similar to Bayes' rule, to reuse RNN/Transformer models that are common in recommender systems (Cho et al., 2014; Hidasi & Karatzoglou, 2018; Ma et al., 2020; Wu et al., 2017; Bai et al., 2019). Correspondingly, we call our model Recurrent Intensity Models (RIMs). Note that while TPPs have been widely discussed in the space of user-churn prediction, item trends prediction, and physical event modeling (Jia & Benson, 2019; Du et al., 2016; Leskovec et al., 2009; Chen et al., 2020; Hosseini et al., 2017; Wang et al., 2019; Li et al., 2021), we believe that TPPs are relatively novel in the UserRec domain, which opens new venues for their applications.

For completeness, while we focus on user-affinity modeling without interventions, our approach can be extended to consider (Granger) causality effects of marketing interventions. In Appendix A, we will discuss another example of UserRec where we aim to send email reminders to users who leave their online-shopping carts abandoned for a period of time (Lowe & Craig, 2021). Different from affinity-based UserRec, the focus there is on the temporal correlations between two event types: the reminder event and the checkout event. Our RIM framework can extend to this case, but we omit the discussion for now, because similar intervention signals may not be easily available in the most general cases. Our current discussion is analogous to implicit-feedback recommendation.

Table 1: Three marketing strategies where recurrent intensity models may apply

|  | UserRec | Offline Match | Online Match |
|---|---|---|---|
| Pick top users for one marketed item in one go | ✓ | ✓ | ✓ |
| Balanced promotion of multiple items at the same time, via constrained optimization over observed user-states |  | ✓ | ✓ |
| Real-time marketing integrated with recommendation, via Dual algorithm with real-time score modifications |  |  | ✓ |

Now that we introduced RIMs for marketing predictions, we continue to discuss some practical challenges in marketing *planning*. Naively, we may simply send push messages to the top UserRec lists to fulfil the marketing campaigns in one go. However, marketing campaigns are often set over a long period of time, which allows more freedom for less intrusive strategies. Table 1 summarizes these new strategies. Particularly, we use OnlnMtch to achieve the exploration goals in our experiments.

To solve OnlnMtch, we develop a *Dual* planner based on the empirical user-state distribution from the past. Our main challenge is to decouple the *one user* whom we need to answer immediately from *all users* whom the marketing campaign targets. We achieve this by finding globally-calibrated *dual variables* to modify the original recommendation scores, based on the theories of optimization duality. Our Dual algorithm is conceptually similar to (Huang et al., 2016; Mehta, 2013; Ding et al., 2019), but we use a different approach via entropic regularization similar to Sinkhorn's algorithm (Cuturi, 2013). We found our variant to be easier to scale and converge in our experiments.

The rest of the paper will show details of our RIM model, Dual planner, and experiments. Additional discussions on related work can be found in the appendix.

## 2 RECURRENT INTENSITY MODELING (RIM)

We are interested in modeling a temporal process represented by a series of event times and types of a given user. Let $\mathcal{S} = [(t_j, y_j) : \forall t_1 < t_2 < \cdots < t_j < \ldots, \forall y_j \in \mathcal{Y}]$ to be a random process that generates user behaviors with a continuous time variable $t$ and a type variable (i.e., the consumed item in the event) in a large-cardinal set $y \in \mathcal{Y}$. We consider a *user-time* as a unique instance variable $X = \mathcal{S}(t) = \{(t_j, y_j) \in \mathcal{S} : t_j < t\}$, with observable user history until time $t$. We use lower-case $x = s(t)$ to indicate its realization and $x \in \mathcal{X}$ to be the set of all possible user-time instances in the recommendation problem. In the ItemRec construct, the sequence models are called upon the arrival of a user event to predict the item choice by a conditional categorical distribution:

$$P(Y = y|x) = \frac{\lambda(x,y)}{\sum_{y' \in \mathcal{Y}} \lambda(x,y')}, \text{ where } \lambda(x,y) = \exp\left(w_y^\top h(x) + b_y + m(x)\right), \quad (1)$$

where $h(\cdot)$ is a sequence encoder function, $w_y$ is a row in the decoder weight, which is also interpreted as the hidden embedding of a specific item, $b_y$ is a global bias in the item choices, and $m(x)$ represents any extra terms that cannot be learned because they cancel between the nominator and the denominator. The $m(x)$ term reminds us that the raw ItemRec scores $\lambda(x, y)$ cannot be used to compare across users. We aim to repurpose $\lambda(x, y)$ for UserRec by associating $m(x)$ with a user prior term that reflects their activeness in the RecSys, an important factor in UserRec.

**Timeless Bayes rule versus temporal intensity** Given our goal to $\text{select}(x|y)$, our first thought is to frame it by Bayes rule $p(x|y) \propto p(y|x)p(x)$, i.e., relating $m(x)$ to $\log p(x)$ and other normalization terms. However, this leads to a paradox shown in Figure 2. The paradox is because we represent a user as a timeless variable $x$, instead of a temporal point process (TPP) through the lens of $s(t)$.

TPP predicts *when* the next event will happen for a user in addition to *what* type it is. TPP is different from other temporal models in that it predicts the event time as a moving target, whereas the other models aggregate time as a feature. The temporal concept is best explained from a counting perspective. Here, we model the absolute number of user-item interactions, $\Delta N(x, y)$, in a future time window, $[t, t + \Delta t)$, where $s(t)$ coincides with $x$. We adopt a Poisson assumption that the rate at which the desired events happen is smooth in time, given by an *intensity* parameter:

$$\lambda(s(t), y) = \lambda(x, y) = \lim_{\Delta t \to 0} \mathbb{E}\left[\frac{\Delta N}{\Delta t}\Big|x, y\right] = \mathbb{E}\left[\frac{\mathrm{d}N}{\mathrm{d}t}\Big|x, y\right]. \quad (2)$$

| Truncated user history in last week ($x$) | $P(y = D\|x)$ |
|---|---|
| $A, B$ | 0.4 |
| $C$ | 0.8 |
| $\emptyset$ | |
| $\emptyset$ | 0.3 |
| $\emptyset$ | |

(Bayesian) $\times \begin{cases} p(x = [A, B]) = 0.2 \\ p(x = C) = 0.2 \\ p(x = \emptyset) = 0.6 \end{cases} \Rightarrow \begin{cases} p(x = [A, B]|y = D) \propto 0.2 \times 0.4 = 0.08 \\ p(x = C|y = D) \propto 0.2 \times 0.8 = 0.16 \\ p(x = \emptyset|y = D) \propto 0.6 \times 0.3 = 0.18 \end{cases}$

(MTPP) $\times \begin{cases} \lambda(x = [A, B]) = 2 \\ \lambda(x = C) = 1 \\ \lambda(x = \emptyset) = \mu \end{cases} \Rightarrow \begin{cases} \lambda(x = [A, B]|y = D) \propto 2 \times 0.4 = 0.8 \\ \lambda(x = C|y = D) \propto 1 \times 0.8 = 0.8 \\ \lambda(x = \emptyset|y = D) \propto \mu \times 0.3 = 0.3\mu \end{cases}$

Figure 2: Bayesian paradox. We start with an ItemRec model $p(y|x)$ that assigns nonzero scores to all users, including cold or inactive users, who are abundant in RecSys due to power-law effects. The Bayes rule incorrectly associates the concentration in quantity with the engagement prior of each user, leading to non-intuitive predictions. Instead, our RIM proposal treats each user as a draw from a MTPP and we use the intensity parameter (defined in Figure 1a) to predict their future engagements. $\mu(> 0)$ is a background intensity assigned to all processes, including inactive users.

The definition of the intensity parameter is visualized in Figure 1. Besides matching the expectations, we may further prescribe $\Delta N(x, y) \sim \text{Poisson}(\lambda(x, y)\Delta t)$ as a parametric distribution, especially in short time windows. The Poisson log-likelihood is:

$$\log P\big(\Delta N(x, y) = n \mid \lambda(x, y)\Delta t\big) = n \log(\lambda(x, y)\Delta t) - \lambda(x, y)\Delta t - \log(n!). \tag{3}$$

The intensity parameter connects three variables of our interests: user sequence $x$, item $y$ and our prediction target $\Delta N(x, y)$ in a future time window $[t, t + \Delta t)$. Slicing $\Delta N(x, y)$ in different directions yields different aspects of the problem. Most importantly, we slice by $\Delta t$ to connect it to the distribution of future event times. We expect an active user to have higher intensities in unit time or, equivalently, shorter intervals between events.

**Temporal Point Processes (TPPs)** We focus on the connection between the intensity parameter $\lambda(x, y) = \lambda(s(t), y)$ and the time $t_*$ for the next interaction with item $y$. Intuitively, we split the time interval until the next event into a large set of infinitesimal windows, i.e., $[t, t_*) = \cup_\tau [\tau, \tau + dt)$, and insert Poisson likelihood with label $n(\tau) = 0$ to all previous windows and $n(\tau_*) = 1$ to the last window. Based on (3), we have

$$\sum\nolimits_{t \leq \tau < t_*} \log P\big(dN(\tau) = n(\tau) \mid \lambda(s(\tau), y)dt\big) = \log\big(\lambda(s(\tau_*), y)dt\big) - \sum\nolimits_{t \leq \tau < t_*} \lambda(s(\tau), y)d\tau$$

$$\stackrel{d\tau \to 0}{\Rightarrow} \quad \log p\big(T_* = t_* \mid t; \lambda(\cdot, y)\big) = \log\big(\lambda(s(t_*), y)\big) - \int_t^{t_*} \lambda(s(\tau), y) \, d\tau. \tag{4}$$

Note that the $dt$ inside the logarithmic term disappears when we convert the probability mass function to probability density function with continuous variable $T_*$. Reinhart (2018); Chen (2016) discuss rigorous derivations as well as additional properties.

Equation 4 is known as the inter-arrival time distribution, which directly formulates a loss function to learn the intensity parameters, without the need to explicitly bucketize time in (2). Equation 4 also implies that the intensity function can be time-varying based on relative history $s(t)$. The numerical integration may be solved by importance sampling (Mei & Eisner, 2017), ODE solvers (Chen et al., 2018), or analytical forms in the case of Hawkes processes, which we use in this paper for simplicity.

**Marked-TPP Decomposition** While TPPs can be built for each event type (i.e., item choice) with cross-channel influences from other event types, this may lead to drastically different models than existing sequential ItemRec models. Instead, we utilize a Marked-TPP decomposition,

$$\lambda(x, y) = \lambda(x)p(y|x), \tag{5}$$

where $p(y|x)$ comes from a regular sequence model (1) and $\lambda(x)$ corresponds to the user-global intensity for all item types. MTPP is the framework of our RIM proposals. It resembles Bayes rule and also considers the user sequence $x = s(t)$ as a point process.

## 3 ESTIMATION OF THE INTENSITY PARAMETERS

While RIM is motivated to estimate the intensity of each user-item pair separately, we find it more convenient to utilize existing sequential ItemRec models. Based on (5), we keep the RNN (or Trans-

former) model for its categorical prediction of the preference *direction*, $p(y|x)$, while introducing separate estimators for the *user-intensity prior*, $\lambda(x)$.

**RNN-Pop** is our simplest model, where we use the length of the user histories in the training sets. This approach works due to a homogeneous intensity assumption, where the number of past events directly correlates with the future event intensity. It also assumes that all user activities are recorded over a comparable period of time, which often holds in practice, because RecSys often truncates the training sets by a fixed time for scalability reasons. RNN-Pop is used in the example in Figure 2.

**RNN-Hawkes** To extend the naive RNN-Pop models, we may break the homogeneity assumption to consider user state changes that affect future event intensity. In this regard, Hawkes process assumes a positive stimulation through past user events and a gradual churn-out effect after a period of inactivity. It models these effects via an exponentially-decaying kernel in the intensity parameters

$$\lambda(s(t)) = \mu + \sum_{j:t_j < t} \sum_{r=1}^{R} \alpha_r \phi_r(t - t_j), \text{ where } \phi_r(\Delta t) = \frac{1}{s_r} \exp\left(-\frac{\Delta t}{s_r}\right) 1_{\Delta t > 0}, \text{ and } \mu, \alpha, s > 0.$$

Here, we extend it with a mixture of $R$ latent kernels with learned coefficients and learn it with *tick* software package Bacry et al. (2017). We design the $R$ kernels to have log-spaced half-lives between $10^{-3}$ and $10 t_{\max}$ where $t_{\max}$ is the largest temporal span in user histories. A large-$s_r$ kernel approximates the RNN-Pop model with a step function after each observation, whereas a small-$s_r$ kernel suggests a fast-diminishing effect, often within a short browsing session.

**RNN-Hawkes-Poisson (RNN-HP)** For long prediction horizons, we may also calibrate Hawkes scores to eliminate the effects of fast-diminishing kernels. In fact, the accurate long-horizon model,

$$\bar{\Lambda}(x, y) \sim \mathbb{E}[\Delta N(x, y)] = \mathbb{E}\left[\int_t^{t+\Delta t} \lambda(\mathcal{S}(\tau), y) \, d\tau \Big| \mathcal{S}(t) = x\right], \tag{6}$$

takes the expectation with respect to stochastic integrals. To calibrate for the long-terms intensity scores, we notice that the Hawkes states $\phi(t)$ are already positive. We thus model the final scores as

$$\bar{\Lambda}(x) = \phi(t) \odot \text{softplus}(w) = \phi(t) \odot \log(1 + \exp(w)). \tag{7}$$

Our additive formulation is inspired by (Mei & Eisner, 2017) and modified by allocating the softplus to each coordinate to improve its numerical stability. Another approach is to replace (6) with over-dispersed distributions like negative binomial (Salinas et al., 2020), which we leave as future work.

**Other Baseline Methods** Practical personalized marketing models are often highly customized, e.g., with profitability or seasonality concerns, and they are not always designed to cover all items in the catalog, which requires training labels for all user-item pairs, including user-item pairs with zero activities in a time period. Despite so, we illustrate the general idea to use explicit recency, frequency, monetization (RFM) features for user segmentation (Fader et al., 2005). Here, frequency and monetary spending in related items are correlated with a customer's future life-time value (CLV) with respect to the given item and so is the time elapsed since last purchase with an inverse relation.

We build a model based on graph-convolutional feature aggregation similar to **GCMC** (Berg et al., 2017), followed by Bayesian-personalized ranking loss (BPR) (Rendle et al., 2009) against sampled negative users and items in the large space of all user-item pairs. Notice that this GCMC model uses labels from an explicit time period, which is inferior to RIM because (a) it has fewer usable labels and (b) it bucketizes the labels instead of preserving their continuous time and order. We attempt a mitigation we call **GCMC-Ex** that creates multiple periods to extract more labels and order information. However, we do notice that perfecting this remedy leads to a temporal integration similar to our TPP derivation (4). More details of this model can be found in Section B.

We also consider matrix factorization (MF) methods, such as **ALS** (Hu et al., 2008) and **LogisticMF** (Johnson, 2014) from Implicit package[1] and **BPR** (Rendle et al., 2009) from LightFM package (Kula, 2015). Notably, LightFM-BPR is limited to sample either users or items, but not both. We include our own BPR model for completeness. Unlike RIM and GCMC, MF methods do not consider temporal features.

---

[1] https://github.com/benfred/implicit

---
**Algorithm 1** Dual planner for online (and offline) matching
---
**Require:** $\lambda_{xy} = \lambda(x, y)$, ideally scaled to $\lambda \leq 1$; user-capacity $\alpha = {}^K/|\mathcal{Y}|$; item-constraint $\beta$;
    user-state distribution $P(\mathcal{X})$ from a past period of time; step-size $\gamma$
**Ensure:** $\hat{v}_y$; real-time modified recommendation for any user-state $x$ by $\text{TopK}\{\lambda_{xy} - \hat{v}_y : y \in \mathcal{Y}\}$
    init $\hat{v}_y \sim \text{Unif}(-1, 0), \forall y \in \mathcal{Y}$
    **for** $k$ in $[0, 1, \ldots, 100]$ **do**
        set $\epsilon = 0.8^k$; sample a batch of user-states $X' \sim P(\mathcal{X})$; compute $\{\lambda_{xy} : \forall x \in X', y \in \mathcal{Y}\}$
        find root $r_x(\hat{\pi}_\epsilon(\lambda_{x:}, u'_x, \hat{v})) = 0$ for every $x \in X'$ given $\hat{v}$; project to $u'_x \geq 0$
        find root $r_y(\hat{\pi}_\epsilon(\lambda_{:y}, u', v'_y)) = 0$ for every $y \in \mathcal{Y}$ given $u'$; project to $v'_y \leq 0$
        take step $\hat{v}_y = (1 - \gamma)\hat{v}_y + \gamma v'_y, \forall y \in \mathcal{Y}$
---

## 4   Online Matching and Dual Planning

So far, we have learned RIM models as extensions from RNN/Transformers using MTPP intuitions. They often yield better accuracy than the time-bucketized RFM baselines and timeless MF models. In this section, we continue to focus on their greater impacts when we consider different marketing scenarios in Table 1. Our main goal is to derive the marketing *dual variables*, $\hat{v}_y$, which directly integrate with the recommendation score outputs on the "Ensure" Line of Algorithm 1.

**OnlnMtch as Constrained Optimization** We start by reformulating offline UserRec by constrained optimization. Say we want to send marketing emails to Top-C users at a given time to promote Item $y$. We represent the space of all user-states at the given time as $\mathcal{X} \ni x$ and rewrite $\lambda_{xy} = \lambda(x, y)$ for convenience. We express the top-user recommendation as binary assignments $\pi_{xy} \in \{0, 1\}, \forall x \in \mathcal{X}$. It is easy and important to verify that $\pi_{xy}$ can be equivalently solved by:

$$\max_{\pi \in \{0,1\}} \sum_{x \in \mathcal{X}} (\lambda_{xy} \pi_{xy}), \text{ s.t. } \sum_{x \in \mathcal{X}} \pi_{xy} \leq C, \text{ assuming } \lambda_{xy} \geq 0. \tag{8}$$

For online marketing or matching, we extend the user-state set $\mathcal{X}$ to a draw from a distribution $P(\mathcal{X})$, which unions all user-states in a period of time by sampling every user at regular time intervals or based on their actual times of interactions, whichever is appropriate for the application. Each sample point is counted as an independent user-state instance. We also replace the summations over $\mathcal{X}$ in (8) with expectations over the probability space $P(\mathcal{X})$. We then solve for the OnlnMtch assignments $\pi_{xy} \in \{0, 1\}, \forall x \in \mathcal{X}, \forall y \in \mathcal{Y}$ with a linearly-relaxed optimization problem:

$$\max_{0 \leq \pi \leq 1} \mathbb{E}_{xy}(\lambda_{xy} \pi_{xy}), \text{ s.t.} \tag{9}$$

$$\begin{cases} \mathbb{E}_y(\pi_{xy}) \leq \alpha, \forall x \in \mathcal{X} & \rightarrow \text{user-capacity constraint with dual variable } u_x; \\ \mathbb{E}_x(\pi_{xy}) \geq \beta, \forall y \in \mathcal{Y} & \rightarrow \text{item minimal-exposure constraint with dual variable } v_y; \\ \mathbb{E}_x(\pi_{xy}) \leq \beta', \forall y \in \mathcal{Y} & \rightarrow \text{item maximal-exposure constraint that rewrites (8)}; \end{cases} \tag{10}$$

where we express the thresholds in relative terms, i.e., $\alpha = {}^K/|\mathcal{Y}|$ for Top-K ItemRec given a user-state $x$ and $\beta' = {}^C/|\mathcal{X}|$ for Top-C UserRec on behalf of an item $y$. Unique to OnlnMtch, we can also insert a minimal-exposure constraint for each item with threshold $\beta, \forall y \in \mathcal{Y}$. Without loss of generality, we keep $\beta$ and omit $\beta'$ to keep the discussions simple.

For the OnlnMtch problem, our challenge is that the item constraints are measured over a distribution of users in a period of time, while OnlnMtch requires real-time recommendation / marketing decisions. This challenge can be met with optimization duality, which we describe next.

**Dual algorithm** We take ideas from optimization duality to rewrite the constraints in a Lagrangian form. I.e., equation 9 (leaving out the omitted $\beta'$ term) is equivalent to

$$\max_\pi \min_{u \geq 0, v \leq 0} L_\epsilon(\pi, u, v) = \mathbb{E}_{xy}(\lambda_{xy} \pi_{xy}) - \mathbb{E}_x(u_x r_x(\pi)) - \mathbb{E}_y(v_y r_y(\pi)) + \epsilon E_{xy}(H(\pi_{xy}))$$

where $\begin{cases} r_x(\pi) = \mathbb{E}_y(\pi_{xy}) - \alpha & \rightarrow \text{residual in the the user-capacity constraint;} \\ r_y(\pi) = \mathbb{E}_x(\pi_{xy}) - \beta & \rightarrow \text{residual in the item minimal-exposure constraint;} \\ H(\pi) = -\pi \log(\pi) - (1 - \pi) \log(1 - \pi) \\ \qquad\qquad\qquad\qquad \rightarrow \text{entropic regularization for } 0 \leq \pi \leq 1 \text{ with annealed } \epsilon > 0. \end{cases}$

Next, we apply the standard trick to define a dual objective by switching the order of operations:

$$\min_{u \geq 0, v \leq 0} \left\{ d_\epsilon(u, v) = \max_\pi L_\epsilon(\pi, u, v) \right\}. \tag{11}$$

After some derivations in Lemma S1 in Section C.1, we find an analytical solution to the inner-loop:

$$\hat{\pi}_{xy} = \text{sigmoid} \left( \frac{\lambda_{xy} - u_x - v_y}{\epsilon} \right); \text{ define as } \hat{\pi}_\epsilon(\lambda_{xy}, u_x, v_y). \tag{12}$$

The solution intuitively suggests that the assignment of a user-item pair not only depends on their predicted intensity score, but also on the global dual variables $u_x$ and $v_y$. For example, if $\lambda_{xy_1} > \lambda_{xy_2}$ for the majority of users while we intend to keep the global exposure rates for $y_1$ and $y_2$ to be equal, then we may choose $v_{y_1} > v_{y_2}$, causing the final item-assignments to be reverted for some users to achieve the desired exposure outcomes. Notice that the item-specific changes $(-v_y)$ are nonnegative, which agrees with our intuition to boost the item exposure rates to at least $\beta$.

Finally, we discuss the solution to the outer-loop, which also yields an intuitive algorithm. To do so, we plug (12) to (11). The derivations in Lemma S2 in Section C.1) lead us to a clean result:

$$\frac{\partial d_\epsilon(u, v)}{\partial u_x} = -r_x(\hat{\pi}_\epsilon(\lambda_{x:}, u_x, v)); \quad \frac{\partial d_\epsilon(u, v)}{\partial v_y} = -r_y(\hat{\pi}_\epsilon(\lambda_{:y}, u, v_y)). \tag{13}$$

Setting both values to zero yields an iterative algorithm that alternates between solving $u$ and $v$. We use simulated annealing from a larger initial $\epsilon > 0$ to produce more stable results. See Algorithm 1 for more details. In the extreme case where $\epsilon \to 0$, the two root-finding steps degenerate as

$$\begin{cases} \text{Find } u_x = \text{TopKThreshold}(\{\lambda_{xy} - \hat{v}_y : y \in \mathcal{Y}\}) \text{ for every } x \in \mathcal{X}; \text{ project to } u_x \geq 0 \\ \text{Find } v_y = \text{Quantile}(\{\lambda_{xy} - u_x : x \in \mathcal{X}\}, \beta) \text{ for every } y \in \mathcal{Y}; \text{ project to } v_y \leq 0 \end{cases} \tag{14}$$

For larger $\epsilon$, we use bisection search to solve for (13) in each step, which is possible because the gradient functions are monotonic with respect to the variables $u_x$ and $v_y$, respectively. The computational complexity is similar to finding quantiles by full sorts. See Appendix C.2 for further discussions on the computational and storage complexity.

**Connections to Online Bidding and Pacing.** Real-time bidding is a common practice in the online advertising industry. The bidding price from an advertiser depends on the expected reward due to the placement, the estimated competitor-pricing distribution, as well as its pacing plan to manage the campaign budget over a period of time. In our Dual algorithm, we may see flavors of all the components: the dual variable $v_y$ affects the overall ranking of the item in each marketing opportunity, receives influences from all other items in each iteration of the alternating algorithm (14), and is calibrated against the empirical user-state distribution to achieve smooth pacing over time. Additionally, Dual planner may extend to future time periods if we assume stationary user-state distribution, which we empirically observe and discuss further in Section F.

Dual planner has direct applications in guaranteed display advertising, where advertisers buy contracts to show their ads to a guaranteed size of audience in a period of time (Bharadwaj et al., 2010). Further, Huang et al. (2016); Mehta (2013); Ding et al. (2019) analyzed the online regrets in such guarantees with slightly different algorithms. These algorithms are designed to minimize online-regrets in adversarial settings. Comparatively, our work is under stochastic and stationary assumptions, but we may consider adjustments after short periods of time for more practicality. Also, we found some difficulties in the convergence of their algorithms, potentially due to the skewed intensity-score distributions in our experiments. We leave the detailed comparisons for future work.

## 5 EXPERIMENTS

We conduct experiments in three RecSys datasets with unique properties.

- **Movielens (ML)** (Harper & Konstan, 2015) is a movie rating website that collects user preferences to study RecSys. We use a small-scale ML-1M in an 2003 release with one million rating events between six thousand users and three thousand items to easily validate our ideas.

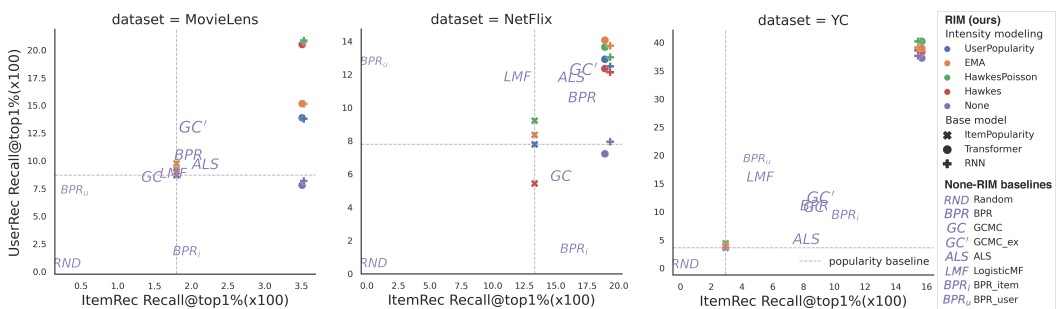

Figure 3: Suitability of different methods for ItemRec and UserRec tasks, plotted on different axes.

- **Netflix (NF)**[2] is a movie rental service where movie ratings from a user's past affects their personalized ItemRec in the future. NF is widely used in rating prediction and ItemRec, but less so in UserRec and OnlnMtch.
- **Yoochoose (YC)**[3] is a dataset used in RecSys 2015 Challenge with six months of activities for product recommendation such as general tools, toys cloths, electronics, and much more.

We highlight some data preparation steps. Most of our RIM methods are self-supervised sequence models. Thus, we only need to hold out test windows with equal horizons for the evaluation of UserRec, ItemRec, and Mtch. We use absolute time windows on NF and consider only the observed users/items in the training data as testing targets, because we may not know which new users/items will show up in the testing windows without temporal leakage from the test set. Besides temporal splits, it is common for sequence models to split train/test sets by users. We consider user-splits on ML and YC, because most users are active at different, non-overlapping clock times. We hold out time windows only on the test users (Group-B in Table S1), with equal starting time relative to their first event as well as equal size or horizon. All training users (Group A) and the observed histories of the testing users (Group B left part) are considered training data.

RNN-HP and GCMC(*) require further splitting of the training set. On NF, we create a set-back window between $[T', T)$ from all users and on ML/YC, we keep the same time $[T, T + \Delta T)$ but change the user base to Group A for validation. Similarly, to simulate OnlnMtch, we must calibrate Online-Dual on different users or times to test the transferability of the user-state distributions in online scenarios. We reuse the set-back window as RNN-HP and GCMC(*) for simplicity. Statistics of our data splits are shown in Table S1.

## 5.1 USERREC AND ITEMREC RESULTS

We first compare various models for the separate tasks of ItemRec and UserRec to validate our proposals in Section 3. On the ItemRec dimension, we observe the usual story that user-conditional models outperform non-personalized models, but they are then topped by sequence models. However, the comparison becomes interesting in the UserRec direction, where the adoption of raw probability scores in RNN and Transformers could lead to inferior results than the simple heuristic that recommends top users unconditionally of the given item, i.e., UserPopularity or the more advanced TPP models for the marginal prediction, $\lambda(x)$. This is because the user-conditional probability scores in RNN/Transformer fail to identify the active users with higher intensity priors. On the other hand, with our RIM proposals, the sequence models become top in the UserRec direction as well.

For the baseline BPR methods, we also compare our implementation with open-source LightFM package. LightFM only allows User/Item sampling, but not both at the same time. In contrast, our version samples both sides concurrently, allowing a single model to perform well in both tasks. We also compare GCMC with GCMC*, a variant that includes multiple time periods to extract more labels and relations. We include similar plots for the precision metrics in the appendix.

---

[2] https://www.kaggle.com/netflix-inc/netflix-prize-data
[3] https://www.kaggle.com/phhasian0710/yoochoose

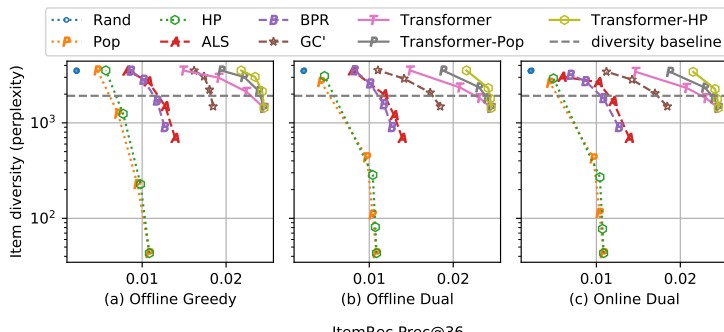

Figure 4: Matching experiments where we set minimal-exposure constraints for every item in additional to regular ItemRec settings. Showing a subset of representative methods on ML dataset.

## 5.2 OFFLINE AND ONLINE MATCHING EXPERIMENTS

To validate the discussions in Section 4, we integrate UserRec into ItemRec by setting minimal-exposure constraints. We fix $\alpha = 1\%$ in equation 9 and vary $0 \leq \beta \leq 1\%$ as item min-exposure constraints. (Section F covers a more common marketing scenario where the user-to-item capacity ratio is greater than one, which leads to even larger margins in our RIM methods. Here, we focus on the case with tighter constraints to highlight the subtleties and new possibilities in our Mtch setting.) Choosing $\beta = 0$ corresponds to the unconstrained ItemRec problem. As we increase $\beta$, we observe an increase in item-exposure diversity, measured by

$$\text{perplexity} = \exp\left(-\sum\nolimits_y \left[\left(\frac{\pi(y)}{\sum_y \pi(y)}\right) \log\left(\frac{\pi(y)}{\sum_y \pi(y)}\right)\right]\right), \tag{15}$$

where $\pi(y)$ is proportional to the total number of times that $y$ is recommended in front of a user.

Figure 4 compares the performance of various methods in three matching environments: Offline-greedy, Offline-Dual and Online-Dual. Within each environment, we see that all methods exhibit trade-offs between ItemRec relevance and item diversity. Comparatively, Transformer-Pop/HP yields best performance and compromises least relevance to achieve better diversity, which supports our claim that better modeling of user intensities leads to better welfare for both users and items. Notice that showing diversity effects in offline experiments is a challenging task due to the vast amount of missing data for alternative treatment effects. We attribute our success to the consideration of new problem dimensions. We show similar results in Section F on other datasets and settings.

We see similar results across different environments in Figure 4(b)&(c). Notably, the Online-Dual solver supports both real-time decisions and long-term exposure constraints over the distribution of all user-states in a future period of time. The consistent performance across different scenarios demonstrates great practicality in the integration of ItemRec and UserRec in RecSys.

## 6 CONCLUSION

RecSys that purely focuses on the immediate user satisfaction may hurt the overall diversity of the items being recommended. Instead, we study an exploratory marketing mechanism where each item will be guaranteed a minimal exposure rate to a suitable audience in a period of time. In this way, we address the pain points in marketing and allow the item creators to focus their expertise on creation. We believe this improves the long-term diversity and liveliness of the RecSys.

We make two contributions. Firstly, we turn a sequential ItemRec model around to recommend users on behalf of a given item. Our challenge is to treat the users as time series instead of random variables. We introduce novel connections to the intensity parameter of a temporal point process and develop a RIM solution that combines RNN (Transformer) and TPP. Our second contribution is to allow a RecSys to guarantee a minimal (maximal) exposure rate for every item that we intend to nourish (downsize) in a future time period. We call this scenario OnlnMtch and introduce Dual planner as a solution. All of our methods learn by user mini-batches to achieve unlimited scalability in the size of the user-state set. Also, compared with other exploration methods that require online feedback, our methods have unique advantages in the ease of benchmarks using offline datasets.

## ACKNOWLEDGEMENTS

We thank Vaibhav Sethi and Balakrishnan (Murali) Narayanaswamy for some early discussions on the idea to repurpose ItemRec models for UserRec applications. We thank Danielle Robinson and Yuyang (Bernie) Wang for their discussions on the topic of Neural-ODEs, which inspired us to look further into the literature around temporal point processes. We appreciate the discussions with Lihong Li and Hsiang-Fu Yu on the general topic of targeted advertising. We finally appreciate the anonymous reviewers for this and our earlier versions for their constructive comments and additional references.

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

## A  RELATED WORK AND DISCUSSIONS

**Non-sequential marketing strategies**  Marketing is a general area studied from multiple perspectives. From a system's perspective, our example of keyword and related-item bidding comes from close examination of large-scale extreme classifier models (Chang et al., 2020). The push notification example considers explicit user states, which are commonly expressed via Recency, Frequency, and Monetary values (RFM) (Fader et al., 2005). By far, the most common and straightforward approach is to use Gradient Boosted Decision Trees to regress on the binary user-item engagement labels using these RFM features. At a simpler level, without the manually constructed temporal features, the loss function is shared with traditional matrix factorization approaches, such as Alternating Least Squares (Hu et al., 2008), Bayesian Personalized Ranking (Rendle et al., 2009), and Logistic Matrix Factorization (Johnson, 2014), all of which serve as UserRec baselines.

**Neural temporal point processes**  We further receive inspirations from temporal point processes (Reinhart, 2018; Chen, 2016; Du et al., 2016; 2015), Neural-ODEs (Chen et al., 2018), and their intersections (Jia & Benson, 2019; Chen et al., 2020; Mei & Eisner, 2017; Zuo et al., 2020). Most of the works are great introduction of various aspects of TPPs and their modern development in GPU environments. However, most of the works focus on the prediction for better temporal modeling without novel applications of the intensity parameters for different applications. Along the line of better temporal models, (Jia & Benson, 2019) shows how Neural-ODE can be applied to a MTPP construction and (Boyd et al., 2020) focuses on the particular challenge that personalized user histories "exhibit large predictive uncertainty at the beginning of the sequences". We do observe that many users churn out shortly after initial engagement. To model this temporal heterogeneity near the users' beginning, we use a simple trick, where we add another related time series with exactly one event for every user to mark their start time.

**Causality**  In our work, we consider item exposure as the primary factor and assume positive downstream effects afterwards. We could also analyze the temporal correlation between an intervention event and the change in the user's temporal intensity in another event type. In the abandoned-cart example, let $\tilde{y}$ be the email-reminder event at time $\tilde{t}$ and $y_*$ be the desired user-checkout event at time $t_*$. We then define the intervention state of the user as $\tilde{x} = x \cup \{(\tilde{t}, \tilde{y})\}$ and model the temporal intensity of the subsequent checkout event by

$$p(T_* = t_*|\tilde{t}; \lambda) = \log(\lambda(\tilde{x}, y_*)) - (t_* - \tilde{t})\lambda(\tilde{x}, y_*), \tag{16}$$

or its time-varying integral form similar to equation 4. Another difference to the likelihood-based intensity models in our main text is the inference routine. To choose which users to send the email reminders, we must also consider the users in the hypothetical intervention states. I.e., we augment the user time series as $\tilde{x} = x \cup \{(\tilde{t}, \tilde{y})\}$ before computing their intensity scores in the effect items.

Besides the (Granger) causality research, temporal intervention effects can also help prevent bad consequences in marketing. For example, we may combine the positive effects of a marketing email through likelihood-based intensity modeling as well as the risks of undesirable effects like user churn-out through causality models. Some interesting work on temporal causality include (Xu et al., 2016; Li et al., 2018; Jiang & Li, 2016; Lundberg & Lee, 2017).

**Convex optimization and online regrets**  On the long-term constraints, we reference from Sinkhorn's algorithm from optimal transport (Cuturi, 2013; Genevay et al., 2018) and the literature in long-term marketing via online campaigns (Huang et al., 2016; Jenatton et al., 2016; Ding et al., 2019), with convergence analysis dating back to (Mehta, 2013). Online convergence analysis further leads to novel applications such as online classification with specificity constraints (Bernstein et al., 2010).

**Diversity and Exploration**  In our discussion of the Dual solution for OnlnMtch, we mention a dual solution in the form:

$$\max_{\pi \in (0,1), r_x(\pi) \leq 0} \mathbb{E}_y[(\lambda(x, y) - v(y))\pi(x, y)] = \text{TopK}\{\lambda(x, y) - v(y) : \lambda(x, y) - v(y)\}.$$

This solution is intrinsically connected to the idea of bandit exploration with upper-confidence bound (UCB) (Li et al., 2010; Chu et al., 2011), where our adoption of the the minimal-exposure constraint

leads to a positive boost to the raw prediction scores, i.e., $-v(y) \geq 0$. Further exploration in this direction may lead to additional benefits in the OnlnMtch paradigm.

# B  GCMC MODEL DETAILS

The basic idea in GCMC-CLV is to split user behavioral data in time to create a temporal-prediction scenario. We hold out a targeting window from the observed user behaviors as CLV labels and build RFM features for every user at the start of the window. Here, every user is represented by the aggregation of the items found in the user's own history, which reflects the frequency of interactions in these items. Due to the sparsity of the user-item pairs in the observed user histories, we use a graph convolution layer for the aggregation, which also extracts the semantic item embeddings similarly to graph-convolutional matrix factorization (GCMC). As for recency, we construct a single user bias term based on the recency of the last purchase regardless of the item choices. We omit the per-item recency terms due to their high (inverse-)correlation with the per-item frequency features. We neither have monetary information in our experiments.

The main challenge in the scalable CLV approach is the vast amount of implicit-negative labels. While we observe positive CLVs in the target window, we should also include every unobserved user-item pair as a zero CLV label. This leads to an imbalanced label distribution, i.e., a significant amount of zeros which we call negatives, and a small portion of non-zeros which we treat as positive (binary) labels. To address the imbalance, we adopt a strategy of negative sampling to augment every positive user-item pair with negative users ($\tilde{u}$) and items ($\tilde{\imath}$), followed by sigmoid-triplet loss:

$$
\begin{aligned}
\text{loss} = 0.5 \Big( & \mathbb{E}_{\tilde{\imath}} \log(\text{sigmoid}(\text{score}(u, \imath) - \text{score}(u, \tilde{\imath})) \\
& + \mathbb{E}_{\tilde{u}} \log(\text{sigmoid}(\text{score}(u, \imath) - \text{score}(\tilde{u}, \imath))) \Big)
\end{aligned}
$$

Details of our negative sampling strategy is similar to Bayesian Pairwise-Ranking (BPR). However, open-source BPR implementations are often limited to sampling only negative items, leaving negative users unconsidered and causing biases in the CLV predictions. We thus include our implementation, with the additional benefit of better integration with GCMC framework. We sample the negative users (and items similarly) according to

$$
p(u) \propto (C(u) + 1)^{0.5},
$$

where $C(u)$ is the total number of appearances of the user (or item) in the training data.

Finally, the BPR scores are translation-invariant, because BPR only focuses on the relative comparisons instead of the absolute values. This would be a problem when we port the scores for Dual, where we expect positive intensity scores. We use softplus activation to preserve the larger values in the raw scores. Our earlier attempt with sigmoid activation leads to saturation in the higher score values, which causes issues due to over-exploration.

# C  DUAL ALGORITHM: ADDITIONAL DETAILS

## C.1  DETAILED PROOFS OF PRIMAL AND DUAL SOLUTIONS

**Lemma S1.** *Show that the solution to* $\max_\pi L_\epsilon(\pi, u, v)$ *is* $\hat{\pi}_{xy} = \text{sigmoid}\left(\frac{\lambda_{xy} - u_x - v_y}{\epsilon}\right), \forall x \in \mathcal{X}, \forall y \in \mathcal{Y}$.

*Proof.* The proof is done by taking derivatives with respect to $\pi_{xy}, \forall x \in \mathcal{X}, \forall y \in \mathcal{Y}$. We first expand out the Lagrangian form

$$
\begin{aligned}
L_\epsilon(\pi, u, v) = \mathbb{E}_{xy}(\lambda_{xy}\pi_{xy}) & - \mathbb{E}_x(u_x(\mathbb{E}_y(\pi_{xy}) - \alpha)) - \mathbb{E}_y(v_y(\mathbb{E}_x(\pi_{xy}) - \beta)) \\
& + \epsilon \mathbb{E}_{xy}(-\pi_{xy} \log(\pi_{xy}) - (1 - \pi_{xy}) \log(1 - \pi_{xy})).
\end{aligned} \tag{17}
$$

Then, we take a derivative with respect to $\pi_{xy}$ as

$$\frac{\partial L_\epsilon(\pi, u, v)}{\partial \pi_{xy}} = \lambda_{xy} - u_x - v_y - \epsilon \log(\pi_{xy}) - \epsilon + \epsilon \log(1 - \pi_{xy}) + \epsilon$$

$$= \lambda_{xy} - u_x - v_y - \epsilon \log\left(\frac{\pi_{xy}}{1 - \pi_{xy}}\right).$$

Setting the derivative to zero yields the desired solution

$$\log\left(\frac{\pi_{xy}}{1 - \pi_{xy}}\right) = \frac{\lambda_{xy} - u_x - v_y}{\epsilon} \quad \Rightarrow \quad \pi_{xy} = \text{sigmoid}\left(\frac{\lambda_{xy} - u_x - v_y}{\epsilon}\right).$$

□

**Lemma S2.** *Show that the solution to $\min_{u \geq 0} d_\epsilon(u, v)$ is in the roots of*

$$0 = -r(\hat{\pi}_\epsilon(\lambda_{x:}, u_x, v)) = \alpha - \mathbb{E}_y(\hat{\pi}_\epsilon(\lambda_{xy} - u_x - v_y)),$$

*projected to $u \geq 0, \forall x \in \mathcal{X}$. The other part of $\min_{v \leq 0} d_\epsilon(u, v)$ can be similarly derived.*

*Proof.* The proof is done by plugging in the primal solution from Lemma S1 to the Lagrangian form and then taking derivatives with respect to $u_x, \forall x \in \mathcal{X}$. We first rearrange the Lagrangian form:

$$L_\epsilon(\pi, u, v) = \mathbb{E}_{xy}(\underline{(\lambda_{xy} - u_x - v_y)\pi_{xy}}) + \alpha \mathbb{E}_x(u_x) + \beta \mathbb{E}_y(v_y)$$

$$+ \mathbb{E}_{xy}\left(\underline{-\epsilon \pi_{xy} \log\left(\frac{\pi_{xy}}{1 - \pi_{xy}}\right)} - \epsilon \log(1 - \pi_{xy})\right)$$

This simplifies the substitution of the solutions from Lemma S1. Specifically, the underlined terms may cancel. Define $z_{xy} = \frac{\lambda_{xy} - u_x - v_y}{\epsilon}$, such that $\hat{\pi}_{xy} = \frac{e^{z_{xy}}}{1 + e^{z_{xy}}}$. The rest of the terms are

$$d_\epsilon(u, v) = \alpha \mathbb{E}_x(u_x) + \beta \mathbb{E}_y(v_y) + \mathbb{E}_{xy}(-\epsilon \log(1 - \hat{\pi}_{xy}))$$

$$= \alpha \mathbb{E}_x(u_x) + \beta \mathbb{E}_y(v_y) + \mathbb{E}_{xy}\left(\epsilon \log\left(1 + e^{z_{xy}}\right)\right).$$

Taking the derivative with respect to $u_x$ leads to

$$\frac{\partial d_\epsilon(u, v)}{\partial u_x} = \alpha + \mathbb{E}_y\left(\epsilon \frac{e^{z_{xy}}}{1 + e^{z_{xy}}} \frac{\partial z_{xy}}{\partial u_x}\right) = \alpha - \mathbb{E}_y\left(\frac{e^{z_{xy}}}{1 + e^{z_{xy}}}\right).$$

Setting the gradients to zero leads to a solution to the unconstrained problem $\min_u d_\epsilon(u, v)$. Additionally, since the gradients are monotone and independent with respect to $u_x, \forall x \in \mathcal{X}$, we may also obtain a solution to the constrained problem in the space of $u \geq 0$ by direct projections. □

### C.2 Complexity and Storage Complexity

With a batch-size of $B$ and an catalog size of $|\mathcal{Y}| = n$, this leads to a temporary storage complexity of $O(Bn)$ and a computational complexity of $O(Bn(\log(n) + \log(B)))$ for each mini-batch. Notice that to enable fast computations on streams of users, we must also avoid the storage of all user-item scores, but use a combination of sparse and low-rank matrices. For example, we may delay the final dot-product layer between the user and item representations in RNN/Transformer models, but instead just store the $D$-dimensional representations using as little as $O((B + n)D)$ storage. The computational complexity for the dot-product layer is $O(BnD)$, which is comparable with sorting / bisection costs in the Dual algorithm. In our experiments, we use $100m/B$ mini-batches based on $m$ total user states in empirical data.

### D Data Statistics and Offline Precision Metrics

Figure/Table S1 shows our data splitting strategies and resulting statistics in the three datasets we consider in Section 5. With temporal split, we carve out a prediction time window between $(T, T + \Delta T]$ and a calibration/validation window between $(T', T]$ for time-bucketed models like Hawkes-Poisson and GCMC and the observation of user-state distribution for Online-Dual simulation. With user split, we pick Group B user to construct a test window between $(T, T + \Delta T]$.

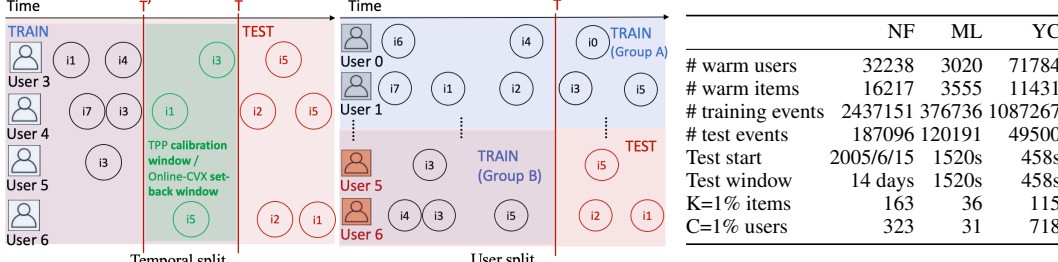

| | NF | ML | YC |
|---|---|---|---|
| # warm users | 32238 | 3020 | 71784 |
| # warm items | 16217 | 3555 | 11431 |
| # training events | 2437151 | 376736 | 1087267 |
| # test events | 187096 | 120191 | 49500 |
| Test start | 2005/6/15 | 1520s | 458s |
| Test window | 14 days | 1520s | 458s |
| K=1% items | 163 | 36 | 115 |
| C=1% users | 323 | 31 | 718 |

Table S1: Data split and statistics

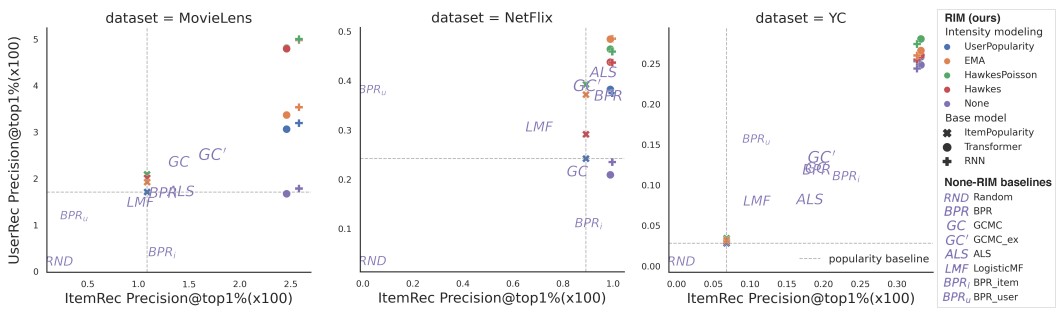

Figure S1: Suitability of different methods for ItemRec and UserRec tasks, plotted on different axes.

For calibration/validation, we use Group A users with the same temporal splits. Our definition of user-state distribution allows sampling by either users or time or both users and time at the same time.

Figure S1 shows the offline precision benchmark results which resemble our Figure 3 on recall metrics in Section 5.1. The trends are similar. We show recall in the main text because it caters more to user-welfare, whereas precision connects to the global objective in Mtch experiments. Conversely, the precision numbers agree with the unconstrained optimization results in Figure 4,S2,S3 and Figure S4,S5,S6 for different combinations of tasks and datasets, respectively.

# E    ITEMREC WITH MIN-EXPOSURE CONSTRAINTS (NF AND YC)

Similar to the Netflix results in the main text, we show two more Mtch experiments on NF (Figure S2) and YC (Figure S3). Comparing the Pareto fronts, we observe RNN-TPP variants > RNN > MF > Non-personalized models, which is consistent with our results on ML dataset.

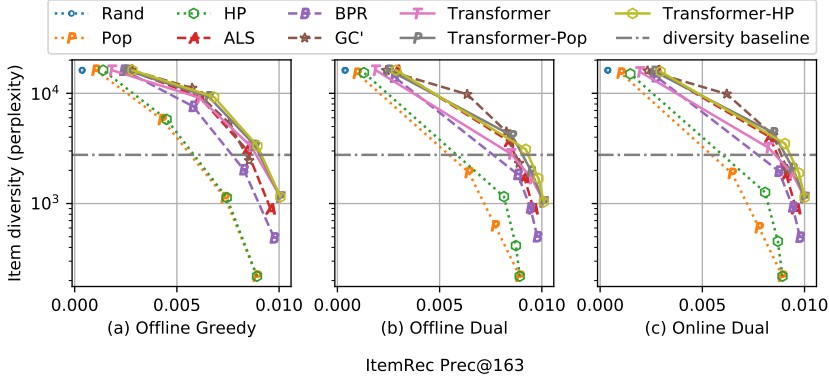

Figure S2: ItemRec with min-exposure constraints in Netflix dataset.

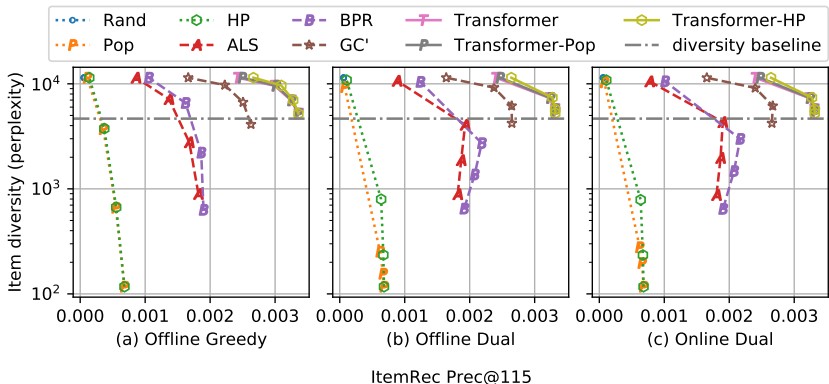

Figure S3: ItemRec with min-exposure constraints in Yoochoose dataset.

# F    USERREC WITH MAX-USER-CAPACITY CONSTRAINTS

We consider $1\% \leq \alpha \leq 100\%$ and $\beta = 1\%$, both as upper-bound constraints in equation 9. In the limit $\alpha = 100\%$, the problem becomes UserRec without the consideration of user capacities. This yields best UserRec relevance, with a risk to overwhelm the users. As $\alpha$ decreases, the users are expected to take a more balanced load of incoming items. We use a similar perplexity metric to measure the user-load balances.

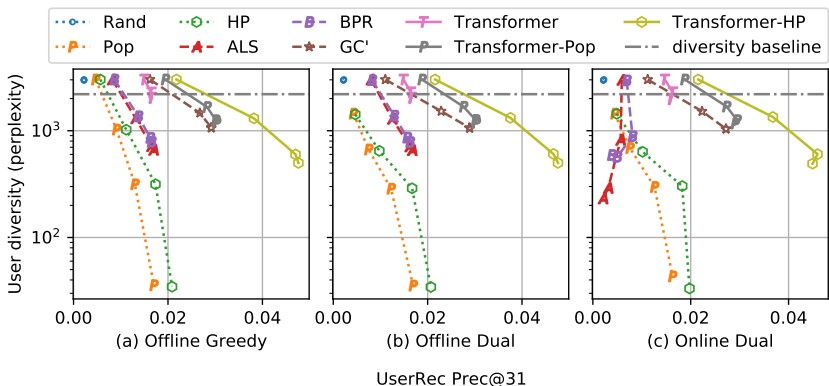

Figure S4: UserRec with user max-capacity constraints in ML-1M dataset.

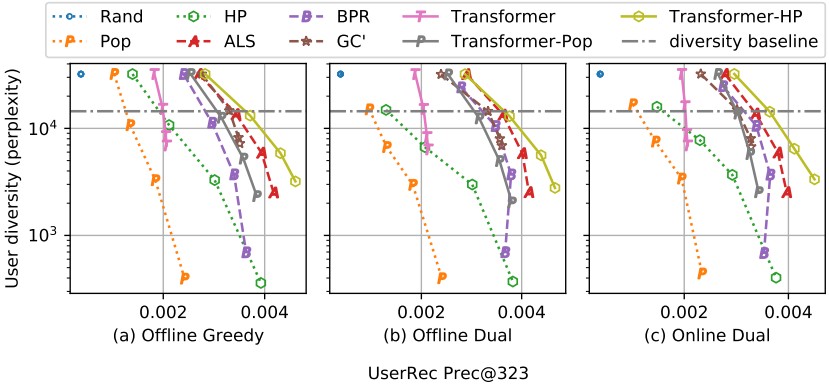

Figure S5: UserRec with user max-capacity constraints in Netflix dataset.

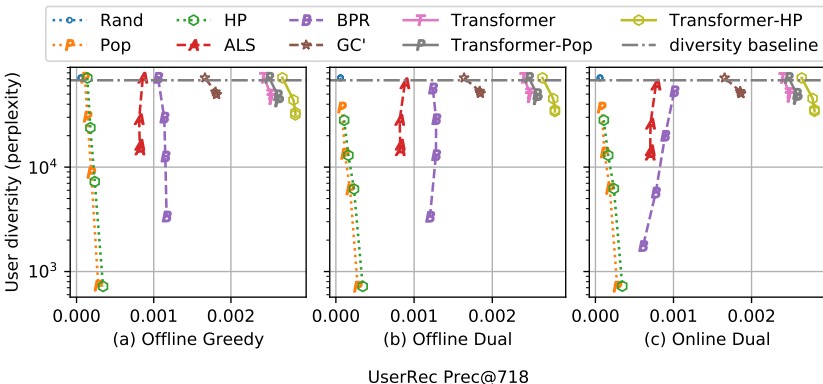

Figure S6: UserRec with user max-capacity constraints in Yoochoose dataset.

Figure S4, S5, S6 show the trade-off between relevance and user diversity, which is an indirect result of capacity limitations per user. From the curves, we find larger advantages in the RNN-TPP variants. This is because the problem is more focused on the advertising perspective, giving larger user-to-item capacity ratio, which highlights our RIM contributions. The rightmost points in the first two plots agree with the unconstrained offline UserRec precision metrics in Figure S1.

For online UserRec, we observe that most methods, including all of RIM and GCMC methods follow similar trends as offline UserRec (Figure S4.c, S5.c, and S6.c). However, we do notice some artifacts in MF methods on ML and YC datasets. The artifacts are related to the under-delivery of recommendations with respect to maximal user-capacity constraints, which leads to drops in the precision metrics. For more details, we show the average percentage of users delivered per marketed item in Figure S7. The ideal percentage is $1\%$ to agree with the $\beta$-constraint. Notice that constraint dissatisfaction is a possibility in online ItemRec as well, but they do not affect our general observation of improvements in diversity-relevance trade-offs and we thus omit the discussions for simplicity. Our discussions here extend to online ItemRec as well.

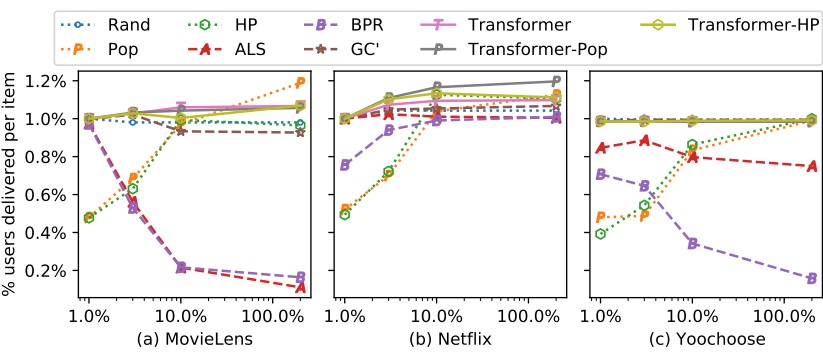

Figure S7: Average percentage of users actually delivered per item. We set $\beta = 1\%$ and $\alpha \geq 1\%$, so the ideal outcome should be $1\%$. However, when we simulate OnlnMtch in a future period of time, any biases in the empirical user-state distribution could cause the outcomes to be different. We observe the bias to be more obvious in ALS and BPR methods on MovieLens and Yoochoose datasets and less so with our RIM and GC methods in all datasets, showing another benefit of stability with sequential models.

What is special in online UserRec is that the total number of recommendations is highly related to the satisfaction of the $\beta$-constraints. For example, in the extreme case where $\alpha = 100\%$ and $\beta = 1\%$, the user capacities are entirely unconstrained, yet the total number of item exposures have to be controlled. The other settings where $1\% \leq \alpha \leq 100\%$ respect the capacity of each individual

user, but still over-provision the overall user capacities for the desired number of item exposures. In these cases, the $\beta$-constraints are observed by transferring the dual parameters, $v(y)$, from the calibration set to the test set through either temporal split (Figure S1.a) or user split (Figure S1.b). I.e., Online-Dual does not use any user statistics in the test set, which are unavailable ahead of time in true online settings. Despite so, the $\beta$-constraints are largely satisfied with RIM and GCMC methods in all datasets as well as MF methods on NF dataset with temporal splits.

The artifacts are only limited to MF methods on ML and YC datasets, both of which use user-splits. Upon further inspection, we realize that the artifacts are due to the fact that we reuse a part of training data for calibration purposes, i.e., the $[T, T + \Delta T)$ time window for Group-A users in Figure S1.b. This causes Group-A users to have more training data, leading to higher recommendation scores during the learning of $v(y)$. As a result, $v(y)$ tends to be higher, causing over-penalization when they are transferred to Group-B users, who have less training data and generally lower recommendation scores. This eventually translates to overall lower item-exposure rates compared with what is calibrated on the validation set, shown in Figure S7. Notice that these artifacts are much less obvious on NF datasets, where the total training-data time (6 months) overwhelms the difference (15 days) between these two splits. They are neither significant in RIM / GCMC methods, where the user training data is averaged against the user time span, improving the stability of the user-state distribution. In fact, the artifacts with MF methods reveal a side benefit of our RIM (and GCMC) methods in our novel design of online experiments.

