# OpenReview forum: "Bridging Recommendation and Marketing via Recurrent Intensity Modeling"
_ICLR.cc/2022/Conference — ICLR 2022 Poster_

### Official Review · Reviewer_cWJc · 2021-11-03

**Correctness:** 3
**Technical Novelty And Significance:** 3
**Empirical Novelty And Significance:** 3
**Recommendation:** 6
**Confidence:** 4

**Main Review:**

Strength:

- The paper attempts to address the user recommendation problem for item marketing, which is relatively underexplored in the RecSys research community but critical in real-world applications.

- The intensity perspective introduced in this paper is technically well-motivated, fresh and interesting.

- The proposed methods were empirically evaluated on three real-world datasets against standard baselines, where the takeaway message is relatively clear. Apart from the user recommendation task, the trajectory perspective also naturally enables user-inductive generalization (i.e., without explicitly learning identifier-based user representation). This property is consistent with the user-inductive experimentation setup in the paper, also close to the real-world online scenarios.

Weakness:

- It is acceptable to me but the position of this paper might be a little bit awkward - it's apparently different from traditional item recommendation scenarios, attempts to address the marketing problem, but doesn't touch the causal aspect or deep dive into the applications such as online advertising or push notifications.

- Another concern is the technical part of this work can be a bit difficult to digest for general RecSys audience. Including running examples might be helpful for readers to grasp the technical ideas at high-level.


**Summary Of The Paper:**

This paper proposes a framework to address the so-called user recommendation (UserRec) problem, tasked to match users (as item consumption sequences) to item products for marketing purposes. The framework extends the Temporal Point Process (TPP) and is built on top of a recurrent intensity model (RIM). The proposed methods were tested on three real-world recommender system datasets against standard baselines on both item and user recommendation tasks, in both offline and online matching experiments.

**Summary Of The Review:**

Although the paper has some caveats and the presentation of the work could use additional improvements, I still find the work interesting and enables a new perspective for the recommender system community. Notable contributions deserve to be called out are 1) it attempts to address the warfare of all parties of multi-sided marketplaces, and 2) it connects user and item utilities in a time-sensitive space. As a researcher working in industry, I'd like to emphasize that addressing the producer/marketer utility is crucial not only for the business success, but also for the market fairness as well as the long-term health of the ecosystem. In this regard, I personally appreciate the contributions of this work.

---

> ### Author Response · Authors · 2021-11-18
> **Thanks**
>
> We appreciate the reviewer’s recognition of our work. Our detailed response is the following:
>
> **[the position of this paper might be a little bit awkward ... causal aspect ... deep dive into applications]**
>
> Causality is possible if we analyze the temporal correlations between an intervention action and the change in hazard rate in a user’s temporal state. While (Granger) causality is a useful consideration in general, e.g., to prevent sending bad emails with large negative effects, it appears more relevant when the marketed item is a discount coupon and the goal is to attract cold or inactive users. We leave this as future work.
>
> We have more results in the appendix that focus on the item advertising perspectives (Figure S4, S5, S6). There we allow a larger user-to-item capacity ratio, which leads to bigger margins with our RIM proposal. We take this away from the main text to focus more on academic discussions and the proposal of new possibilities. We added this comment to the main text as well.
>
> **[Including running examples]**
>
> We added an example we call Bayesian paradox at Figure 2 to showcase the motivation of RIM and some intuitions about why it is needed in UserRec usecase.
>
> **[Notable contributions ... welfare of all parties ... utilities in a time-sensitive space]**
>
> We appreciate the reviewer’s recognitions. We added a discussion in the appendix to suggest how our Online-CVX dual algorithm may contain an interesting connection to the bandit exploration algorithms as future work. We do notice that bandit often requires explicit feedback, yet our MTPP model allows the conversion of implicit feedback via intensity modeling.

---

### Official Review · Reviewer_9NaL · 2021-11-07

**Correctness:** 4
**Technical Novelty And Significance:** 4
**Empirical Novelty And Significance:** 2
**Recommendation:** 8
**Confidence:** 3

**Main Review:**

This is a strong paper with many interesting novel contributions. The problem of UserRec is very naturally modeled by the author's proposed framework and their empirical results are impressive.

My main criticism is that the paper was quite terse and hard to read on a first pass. The paper uses tools that are likely to be unfamiliar to many in the ML community with relatively little introduction. I think this could likely be remedied by moving some of the details that aren't core to the paper to the appendix, and spending more time on exposition during the main body of the paper.

Furthermore, in section 5.1 the authors compare against classic matrix factorization techniques such as ALS. However, they do not compare against matrix factorization methods that take into account the temporal nature of the task, which have been shown to work well in these settings. e.g. timesvd++: https://www.cc.gatech.edu/~zha/CSE8801/CF/kdd-fp074-koren.pdf it would be interesting to see a comparison with such methods.

**Summary Of The Paper:**

This paper considers the problem of recommendation in a temporal context. In this setting, which the authors call UserRec, the recommender system must determine whether to recommend an item to a user at a specific point in time given their prior interaction history, or whether to delay the interaction. They draw connections to classical item recommendation, where the recommender must instead recommend some item to a given user, and does not have the option of delaying the recommendation. By modelling the UserRec problem as a temporal point process (TPP) the authors propose a framework to convert algorithms developed for ItemRec into a UserRec model.

They then propose an algorithm for recommendation in matching that leverages their TPP representation to setup a convex optimization problem. Finally, the authors validate their results on three different datasets.

**Summary Of The Review:**

The paper in its current form is already a strong contribution, however I would appreciate it if the authors focused on exposition more in the final version of their paper.

---

> ### Author Response · Authors · 2021-11-18
> **Thanks**
>
> We appreciate the reviewer’s recognition of our work. We take the feedback for a simplification of our abstract in the revised paper. We also simplified Section 2.1 and added Figure 2 to show a numerical example.
>
> We are not quite able to adapt TimeSVD++ because it is based on explicit labels, whereas we treat all unobserved user-item pairs as implicit negatives. However, in its spirit, we added a baseline using graph-convolutional aggregation of recency, frequency, and monetization (RFM) features, followed by Bayesian personalized ranking loss. This GCMC model is time-bucketed and we build connections to our RIM proposals by taking temporal integrals instead of explicit time windows.

---

### Official Review · Reviewer_CMef · 2021-11-07

**Correctness:** 3
**Technical Novelty And Significance:** 2
**Empirical Novelty And Significance:** 2
**Recommendation:** 5
**Confidence:** 4

**Main Review:**

***Strong aspects***
1. The research problem is well motivated. Item recommendation and item marketing are indeed two important and close-related problems.
2. Modeling users' behaviors by a temporal process makes sense, and it has been demonstrated effective in many works.
3. This paper evaluates the approach on three benchmark datasets.

***Weak aspects***
1. Baselines are weak and old. Competitive sequential models in recommender systems such as the Transformer-based one, SASRec [1], should be considered. Besides, there is no baseline for item marketing.
2. Experimental setup in Table 1 is strange. It is obvious that RNN will have better performance since all of the other methods cannot leverage temporal information.
3. The evaluation is not well explained. Why is diversity needed in the ItemRec task?
4. Paper writing requires improvement. The figures are not so clear, such as Figure 3. Even some basic explanations, such as the difference between UserRec and ItemRec, are not well presented.

[1] Kang, W. C., & McAuley, J.  Self-attentive sequential recommendation. In IEEE International Conference on Data Mining, 2018.

**Update after rebuttal**
The authors have actively provided feedback, and I partly agree with their opinions about baselines. Therefore I have raised my score.

**Summary Of The Paper:**

This paper studies item recommendation (ItemRec) and item-marketing (UserRec) as two variants of the same problem. Specifically, the authors convert ItemRec to UserRec based on the temporal point process with the proposed RIM. The authors evaluate the approach on three widely-used datasets.

**Summary Of The Review:**

Overall the paper is not written well. The studied problem is important, but the proposed method is not well evaluated and cannot be claimed as a good solution to the research problem. In the current review stage, I would like to recommend rejecting this paper, waiting for author feedback to check whether my concerns can be addressed or not.

---

> ### Author Response · Authors · 2021-11-18
> **Addressing concerns (Part 1)**
>
> We appreciate the reviewer’s constructive criticism about our presentation of results, coupled with a potential confusion in the difference between temporal point processes and temporal feature aggregation. We revised our paper accordingly. Here are some detailed responses:
>
> **[Modeling users’ behaviors by a temporal process ... has been demonstrated effective in many works]**
>
> The reviewer seems to refer to Transformer models. Our RIM proposal is different in that we predict time as a moving target, whereas the Transformer model uses time as a context. This is important because to be successful in UserRec, we must predict not only what but also when a user may visit an item.
>
> Our contribution is the proposal of the use of TPP-based user intensity modeling in combination with a base model that predicts conditional distribution, and that the base model can be anything (including Transformer). We demonstrate additional result where Transformer is used as the base model and show that RIM is able to further improve its performance on UserRec because of the capability to predict time.
>
> **[There is no baseline for item marketing]**
>
> Not exactly. The reviewer may appreciate our baselines more, if they realize the inadmissibility of many seemingly good ItemRec models, such as vanilla RNN/Transformer, due to the lack of user-intensity priors. Our UserRec results (Figure 4 in revised paper) show a convincing story about how our RIM proposal leads to better user recommendation in unconstrained marketing settings on behalf of the item providers. In Appendix F, we further consider user-capacity constraints and highlight the contribution of our RIM proposal when there are more user capacities than item capacities. Our main text focuses on the more interesting scenario when the capacity constraints are tight, shown in Figure 5.
>
> To go one step further, we added new baselines in this revision, which directly build from common marketing practices to use recency, frequency, monetization (RFM) features. Notice that the RFM approaches are often highly customized for specific items and profit margins and they are not always designed to cover all items in the catalog, which requires training labels for all user-item pairs, including user-item pairs with zero activities in a time period. For illustrative purposes, our new implementation uses a graph-convolutional network for feature aggregation, followed by Bayesian pairwise ranking against sampled negative users/items as loss function.
>
> What we find is that the usual marketing approaches, which obtain their labels from an explicit time window, often perform worse than RIM because (a) they have fewer usable labels and (b) they bucketize the labels instead of preserving their continuous time and order. We attempted a mitigation by creating multiple windows to extract more labels for GCMC training. However, this is just another way to arrive at our RIM model, which we explain in Equation 4.
>
> **[Table 1 ... obvious that RNN will have better performance]**
>
> We respectfully disagree. To make our points, we changed Table 1 to Figure 4 for direct visual comparisons. While we see the usual trend that RNN / Transformer performs better in ItemRec, the raw RNN /Transformer probabilities are unusable for UserRec - they lead to worse performance than the simple heuristic to always recommend the top user with most histories. The RNN-UserRec performance can be fixed by introducing user-intensity priors, but this is only possible due to our contribution.
>
> **[The evaluation is not well explained. Why is diversity needed in the ItemRec task]**
>
> Diversity is a general topic in RecSys community, though often manifested in different ways. In our proposal, we consider the benefits of both users and items. The user benefits are measured by the relevance in their Top-K recommendations. The benefits of the item providers are included with a promise of a minimal number of exposures for their items. We find that the benefits of users and items can be shared if we frame recommendation as a matching problem with these constraints included. A stronger minimal-exposure constraint leads to more considerations of the welfare of item providers, which also yields larger overall item diversity in the final recommendations, which we show on the vertical axis of Figure 5.
>
> Because we approach diversity from a shared-benefit perspective, we observe that our RIM proposals lead to better overall performance with smaller compromises in recommendation relevance. This is a significant improvement over baseline methods (e.g., vanilla Transformer) that do not consider the welfare of item providers.
>
> We further make some connections to bandit exploration in the appendix, though we want to highlight a key difference that our models use implicit-feedback datasets thanks to our novel MTPP interpretations.

---

> > ### Author Response · Authors · 2021-11-18
> > **Addressing concerns (Part 2)**
> >
> > **[The figures are not so clear, such as Figure 3]**
> >
> > We removed some methods from Figure 3 (now Figure 5) to make the plots less crowded.
> >
> > **[waiting for author feedback to check whether my concerns can be addressed or not]**
> >
> > We thank the reviewer for the flexibility and we hope that we have addressed most of the reviewer’s concern as we highly value the reviewer’s comments. We’re happy to response to any additional questions if available.

---

> > > ### Comment · Reviewer_CMef · 2021-11-29
> > > **Reply to authors' feedback**
> > >
> > > Thanks for the authors' feedback. I have also read the revised paper.
> > >
> > > I partly agree with the authors' replies to my first two questions. As for my third concern of diversity, it is still not so convincing since accuracy is also the most important one.
> > >
> > > Anyway, my issues are partly addressed, and I will raise my score.

---

> > > > ### Author Response · Authors · 2021-11-29
> > > > **Answers to Remaining Questions**
> > > >
> > > > We appreciate the reviewer’s agreement with our contributions of intensity-based recommender models, choices of baselines, and experimental designs. We’d like to provide further clarification regarding diversity (and accuracy) as this is the only remaining concern.
> > > >
> > > > First of all, we showed that our RIM proposals achieve best accuracy in ItemRec and UserRec problems simultaneously (Figure 4 both axes). Additionally, we showed better accuracy when ItemRec and UserRec are combined into a matching scenario, i.e., ItemRec with item-exposure constraints. This novel diversity perspective provides additional insights and impacts, because:
> > > >
> > > > 1. Diversity is a commonly seen constraint in recommender/advertisement systems. We showed an example of display guarantee, where advertisers want to make sure that they surface their new contents to sufficiently many people many times for best effects. In practice, there may be other diversity constraints with respect to item owners, features, freshness, etc. Our matching proposal satisfies such constraints with consideration of the benefits of all users and items at the same time. Interestingly, our diversity pursuits also connect in principle to the contextual-bandit technique that is often used in item cold-start exploration. We included this discussion in our related work section in the appendix.
> > > > 2. We propose a complementary contribution in Online-CVX to complete the missing puzzle about how real-time RIM models can be used in real-time decision environments like ItemRec. This resolves the challenge that any such display guarantees can only be validated in hindsight, long after the decisions are made.
> > > > 3. The mixed user-item perspective reveals that despite people’s best efforts in building the most positive user experience, an actual recommender system may inadvertently be influenced by the marketing/exploration values of the items, benefiting the item providers - in fact, the two-sided system - in return.
> > > >
> > > > Lastly, we found that our major contribution - the intensity-based recommendation models and the UserRec problem - is largely appreciated by all of the reviewers. We believe that this contribution alone suffices the standard of a good paper and thus we sincerely hope that there could be a further adjustment of score with the above discussions that address reviewer CMef’s concern on diversity.

---

### Author Response · Authors · 2021-11-18
**New Revision**

We appreciate the reviewers’ constructive comments and encouragements in our research. To address the major comments from the reviewers, we improved our experiment by adding two additional methods: Transformer (as a new RIM-base model) and GCMC (as a new baseline for comparison), and made some modifications to improve the clarity of the paper. Key changes include:

1. Introduce a Bayesian paradox (Figure 2) to illustrate the role of intensity predictions (Reviewer cWJc)
2. Discuss the difference between TPP and other sequence models in Section 2.1 (Reviewer CMef)
3. Illustrate common marketing practices by implementing a new baseline through graph-convolutional neural network (Reviewer CMef, 9NaL)
4. Better discussion of our first experiment with scatter plot visualization of model performance (Reviewer CMef)
5. Discuss intuitions for diversity improvements in Section 4 (Reviewer CMef)
6. Better visualization of our second experiment by removing clutters (Reviewer CMef)
7. Add Transformer-based models to evaluation benchmarks (Reviewer CMef)

---

> ### Author Response · Authors · 2021-11-21
> **Minor Revision in the Appendix**
>
> Dear Reviewers,
>
> We just wanted to add a minor revision before the paper revision deadline. In this revision, we included some detailed discussions about an interesting observation on the matrix factorization baseline methods in some of our experiments. They are located at the end of the appendix and colored purple. They are completely optional to review.
>
> We also wanted to check back on any further comments regarding our previous revision, submitted on Thursday. We would love to hear your thoughts before the discussion period ends on Monday.
>
> Thanks

---

### Decision · Program_Chairs · 2022-01-20

**Decision:**

Accept (Poster)

**Comment:**

Between a reject, an accept, and a borderline-accept, this is truly a borderline paper, though I'd lean slightly on the side of accepting it. The most negative review raises issues of weak baselines, along with several more minor issues. The authors rebut this reasonably well, arguing several differences from the setting used in the suggested baseline papers. It is a little hard to follow who is correct between the review and the rebuttal, though the rebuttal makes reasonably convincing arguments. Other than the baselines most of the issues raised by the negative review are more minor and can be easily fixed in a revision. The specific issues mentioned are mostly not raised in either of the more positive reviews.

The borderline (positive) review is by far the most detailed, but overall praises the paper and mostly suggests fixes in terms of better positioning the paper. Overall both the positive and borderline-positive reviews make persuasive arguments as to the paper's conceptual merits, which outweigh some more minor issues.